# Language-Conditioned Path Planning

**Amber Xie**[1]      **Youngwoon Lee**[1]      **Pieter Abbeel**[1]      **Stephen James**[2]

[1]University of California, Berkeley      [2]Dyson Robot Learning Lab

https://amberxie88.github.io/lapp/

**Abstract:** Contact is at the core of robotic manipulation. At times, it is desired (e.g. manipulation and grasping), and at times, it is harmful (e.g. when avoiding obstacles). However, traditional path planning algorithms focus solely on collision-free paths, limiting their applicability in contact-rich tasks. To address this limitation, we propose the domain of Language-Conditioned Path Planning, where contact-awareness is incorporated into the path planning problem. As a first step in this domain, we propose Language-Conditioned Collision Functions (LACO), a novel approach that learns a collision function using only a single-view image, language prompt, and robot configuration. LACO predicts collisions between the robot and the environment, enabling flexible, conditional path planning without the need for manual object annotations, point cloud data, or ground-truth object meshes. In both simulation and the real world, we demonstrate that LACO can facilitate complex, nuanced path plans that allow for interaction with objects that are safe to collide, rather than prohibiting any collision.

**Keywords:** Robotic Manipulation, Path Planning, Collision Avoidance, Learned Collision Function

## 1 Introduction

Collision checking is a fundamental aspect of path planning in robotics [1, 2, 3, 4, 5, 6, 7], aiming to find a path between initial and target robot configurations that avoids collisions with the environment. However, traditional collision-free path planning approaches fall short in scenarios where contact with the environment is necessary, such as when manipulating objects or interacting with the surroundings. In such cases, the strict "collision-free" constraint becomes impractical and inhibits the robot's ability to perform tasks effectively.

Traditional approaches for enabling contact in path planning [8, 9] often require manual adjustments, such as disabling collision checking for specific objects. However, these approaches rely on access to object state information, ground-truth object meshes, or extensive engineering efforts for each execution. This poses significant challenges, particularly in vision-based contact-rich robotic manipulation tasks.

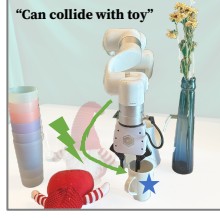

(a) w/o language          (b) w/ language

Figure 1: Language-conditioned path planning enables finding a path, e.g., toward a cup, with acceptable collisions, e.g., a plush toy **(right)**, whereas typical path planning fails at finding a collision-free path **(left)**.

To overcome this limitation, we propose the domain of **Language-Conditioned Path Planning (LAPP)**, which integrates contact-awareness into the path planning problem. In this domain, path planning is not solely concerned with avoiding collisions but also incorporates the ability to make informed decisions about contact with the environment. This enables robots to perform complex manipulation tasks that involve controlled interactions, such as holding a cup or opening a door. Figure 1 provides an illustration of a typical scenario where a robot encounters multiple obstacles and needs to interact with the environment.

7th Conference on Robot Learning (CoRL 2023), Atlanta, USA.

To facilitate flexible and adaptive contact-aware path planning, we propose **Language-Conditioned Collision Functions (LACO)** as an initial step in the language-conditioned path planning domain. LACO learns to predict a collision based on a single-view image, language prompt, and robot configuration. By predicting collisions between the robot and the environment modulated by the language prompt, LACO enables the generation of path plans that can handle both desired and controlled collisions without requiring manual object annotations, point cloud data, or ground-truth object meshes. This approach empowers robots to interact with objects that are safe to collide, rather than rigidly avoiding all collisions.

In summary, our main contributions are threefold:

- We propose a novel domain of Language-Conditioned Path Planning (LAPP) that integrates semantic language commands to enhance the robot's understanding of how and what to interact with in the environment. By fusing language instructions, we enable more intelligent and context-aware planning.

- To enable language-conditioned path planning, we introduce Language-Conditioned Collision Functions (LACO), a collision function that incorporates language prompts to modulate collision predictions. LACO utilizes only a single-view camera, eliminating the need for object states or point clouds. This allows for easier application in real-world scenarios and facilitates zero-shot generalization to new language commands.

- We provide comprehensive demonstrations of LACO's effectiveness in various path planning tasks. Our experiments include simulations and real-world scenarios, highlighting the practicality and robustness of our language-conditioned collision function.

## 2  Related Work

**Path planning** [1, 2, 3, 4, 5, 6, 7] finds a collision-free path between initial and target task configurations – representing robot states and potentially environment states – by querying a collision function with states along a path. However, these collision-free path planning methods struggle at handling scenarios where collisions with the environment is desired. In this paper, we propose language-conditioned path planning, which finds a contact-aware path following a language prompt.

**Semantic planning** [10, 11, 12] builds a semantic map with a specifically-designed perception pipeline, consisting of, e.g., object detection, segmentation, and keypoint estimation modules. The semantic map is then used to find a collision-free path for navigation. In contrast, we propose an end-to-end language-conditioned collision function, which enables contact-aware path planning.

The recent success in **large language models** enables training multi-task robotic policies guided by a language instruction [13, 14, 15] and high-level planning using a large language model [16, 17]. These language-conditioned "task" planning / solving approaches differ from our proposed language-conditioned "path" planning domain in that they directly solve a specific set of tasks while our language-conditioned path planning is agnostic to downstream tasks and can be used as a building block for many robotics tasks.

A **collision function** is a fundamental component of path planning in robotics. However, the collision function is assumed to be manually modelled or computed using state estimation. Instead of hand-engineering a collision function, many recent work have learned end-to-end motion planners [18, 19, 20] or collision functions [21, 22, 23] from synthetically generated data in simulation. To understand 3D configuration of environments, these approaches use point clouds to represent the scenes. In this paper, we propose to learn a collision function from only a single camera input, not requiring depth sensing nor precise camera calibration, which makes our method easily applied to many real-world applications. More importantly, our learned collision function conditions on language describing which objects to collide or not to collide, allowing acceptable or desired collisions for path planning.

Figure 2: A language-conditioned collision function (LACO), $C(o, s, l)$, predicts whether a robot in a state $s$ collides with objects other than collidable objects described in a language prompt $l$ in a scene $o$, e.g., $C(o, s, \text{'pringles'}) = 1$. To find a language-conditioned path plan, a path planning algorithm asks LACO whether any waypoint $s_i$ of a path collides with objects except for the ones described in $l$.

# 3 Language-Conditioned Path Planning

In a cluttered real-world environment, a collision-free path can be highly sub-optimal or impossible to find. We introduce the problem domain of language-conditioned path planning, which extends traditional path planning to allow safe or desired collisions in a path via language, in Section 3.1. Furthermore, we present a proof-of-concept framework for language-conditioned path planning that first learns a language-conditioned collision function (Section 3.2) and leverages this learned collision function for language-conditioned path planning (Section 3.3).

## 3.1 Language-Conditioned Path Planning (LAPP)

In robotics, a path planning problem is about finding a connected, collision-free path of robot configurations (i.e. waypoints), $\mathbf{p} = (s_0, s_1, \ldots, s_g)$, starting from an initial configuration $s_0$ to a final configuration $s_g$ such that every configuration in the path has no collision with the environment, $\forall s_t \in \mathbf{p}, C(o_t, s_t) = 0$. $o_t$ and $s_t$ denote the environment and robot configurations at time $t$, respectively, and $C(o, s)$ denotes a collision function that outputs 1 if the robot configuration $s$ has collision with the environment $o$, and 0, otherwise.

In this paper, we propose a *Language-Conditioned Path Planning (LAPP)* problem, which relieves the strict collision-free constraint in path planning so that path planning can manage safe or desired contacts with the world, *especially guided by language*. LAPP can be formulated as finding a path $\mathbf{p}$ between two configurations, $s_0$ and $s_g$, such that $\forall s_t \in \mathbf{p}, C(o_t, s_t, l) = 0$, where $l$ is a language prompt modulating what needs to be considered as acceptable or desired collisions. For example, a language prompt $l$ can be "a robot can collide with plush toys" to specify safe-to-collide objects, as illustrated in Figure 1, or "a robot can grasp a mug" to support contact-rich tasks.

## 3.2 Language-Conditioned Collision Function (LACO)

For language-conditioned path planning, a path planning algorithm should understand which collisions are acceptable and which must be avoided described in language. In this paper, we address this problem by adapting a collision function $C(o, s)$ into a *Language-Conditioned Collision Function (LACO)* $C(o, s, l)$, which takes a language prompt $l$ into account. Specifically, LACO learns a collision function $C(o, s, l)$, where $o$ is a single-view image observation of the environment, $s$ is a queried robot joint state, and $l$ is a language instruction corresponding to objects that allow for collisions. Note that $o$ does not need to correspond to $s$ and represents only the environment configuration. Thus, the same $o$ can be used across multiple robot configurations $s$.

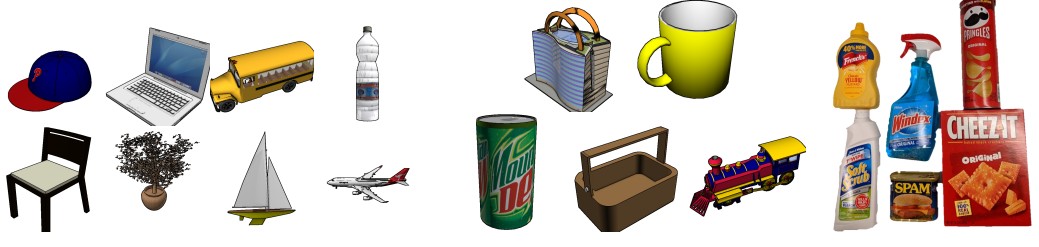

| (a) Training objects | (b) Held-out objects | (c) Real-world objects |

Figure 3: We use ShapeNet objects [27] for our simulated experiments. (a) The training dataset of ShapeNet classes includes: airplane, chair, pot, vessel, laptop, bus, cap, and bottle. (b) The held-out evaluation dataset includes: basket, mug, train, bag, and can. (c) We also perform real-world experiments with YCB objects [28]: Spam, Cheez-it, Pringles, Windex, mustard, and bleach.

We train $C(o, s, l)$ on a dataset $\mathcal{D} = \{(o, s, l, y^l, y)\}$, where $y$ indicates whether the robot state $s$ has any collision in the scene $o$; and $y^l$ indicates whether there is a undesirable contact under the language instruction $l$. We optimize the cross-entropy losses for language-modulated collision prediction (target $y^l$) and collision prediction without language conditioning (target $y$) as an auxiliary task:

$$\mathcal{L} = \mathbb{E}_{(o,s,l,y^l,y)\sim\mathcal{D}} \left[ CE\big(y^l, C(o, s, l)\big) + CE\big(y, C_{\text{aux}}(o, s, l)\big) \right], \tag{1}$$

where $C_{\text{aux}}(o, s, l)$ is an additional MLP head attached to the last layer of $C(o, s, l)$.

To take advantage of large vision-language models pretrained on a large corpus [24, 25], LACO uses the vision and language encoders of CLIP [26] as the backbone networks. As illustrated in Figure 2, we tokenize an input image $o$ of size $256 \times 256$ with the frozen pretrained CLIP ViT encoder and a language prompt $l$ with the frozen pretrained CLIP language model. We then get 197 visual tokens $z^v_{0:196}$ and 197 language tokens $z^l_{0:196}$. For a robot state $s$, we use a 3-layer MLP to embed it to a single state token $z^s$. All these tokens are then fed into a 2-layer transformer and the average of the transformer output tokens is used to predict the collision probabilities with two separate MLP heads, $C(o, s, l)$ and $C_{\text{aux}}(o, s, l)$. More hyperparameters are described in Appendix, Table 7.

### 3.3 Path Planning using LACO

Finally, language-conditioned path planning can be performed by simply replacing a collision checker in any path planning algorithm with LACO. In this paper, we implement LAPP using an optimization-based method, LAPP-TrajOpt [7]. Whenever LACO needs a binary output (collision or not), we apply a threshold of $0.5$ to the collision probability output of LACO. LACO is flexible to various styles of path planning algorithms, such as sampling-based planning. We use a custom implementation of TrajOpt [7] and the hyperparameters for TrajOpt can be found in Appendix, Table 8.

## 4 Experiments

In this paper, we introduce language-conditioned path planning (LAPP), and present a framework that combines existing path planning algorithms and our proposed language-conditioned collision function (LACO) as an initial step. Our evaluations are twofold: (1) we present a thorough investigation of LACO's performance in object- and language-level generalization, and (2) we showcase the potential of LACO in the language-conditioned path planning domain.

### 4.1 Environment Setups

We use the UFACTORY xArm7, a low-cost 7-DOF robotic arm, and an Intel RealSense D435 camera. For the real world, we use 6 YCB objects [28]: Spam, Cheez-it, Pringles, Windex, mustard, and bleach. For simulation, we use ShapeNet v2 objects in CoppeliaSim [27, 29], which provides a taxonomy of diverse, realistic set of 3D meshes with their labels, as shown in Figure 3.

## 4.2 Data Collection

To train LACO, we first collect data in both the simulation and real-world environments.

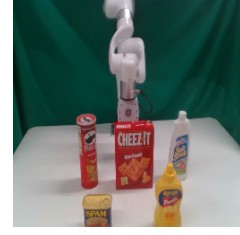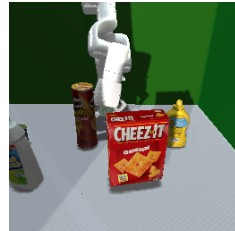

**Simulation dataset.** We use PyRep [30] based on CoppeliaSim [31] to synthetically generate a diverse, language-annotated dataset in simulation. Each scene includes 2-5 randomly chosen objects in random poses on the table. Instead of randomly initializing a robot pose, a set of robot poses are generated by the built-in RRT* motion planner [6] for each scene. These smooth trajectories bias the dataset to-

Figure 4: The real-world (left) and simulated (right) environments.

ward joint states likely to be queried by a path planner. We use the built-in collision checker for the ground truth collision label $y$. Next, we sample combinations of $0, .., N-1$ objects in a scene with $N$ objects to generate language annotations (a list of ShapeNet names of the sampled objects) and compute language-conditioned collisions $y^l$. We generated 5000 unique scenes, which consist of unique combinations and positions of objects. Each scene contains about 40 joint states and 10 language annotations.

**Real-world dataset.** Learning a collision function from real-world data poses additional challenges: the increased visual complexity and the difficulty of collecting collision data. To address these issues, we train LACO on a dataset collected from a domain-randomized twin simulator environment (Figure 4) and then finetune on a small real-world dataset. We collect data from 20 real-world scenes and 500 domain-randomized simulation scenes. For each scene, we extract 20-30 images with domain randomization in simulation and camera perturbations in the real world. Similar to our simulation dataset, we vary the number of objects in each scene from 3 to 5 and vary positions of objects.

## 4.3 Collision Prediction Results Across Language Conditioning

LACO possesses the ability to be modulated by language. In particular, any number of objects in the scene can be included in the language condition, allowing for flexibility in path planning. This ability is not found in built-in collision checkers or learned collision checkers, such as SceneCollisionNet [21], which are agnostic to desired and undesired collisions.

Table 1: We evaluate the language-conditioned collision prediction accuracy of LACO in simulation. For reference, we also evaluate SceneCollisionNet [21], which does not support language conditioning and "Built-In Collision" refers to the ground-truth collision checker.

| Method | Accuracy Per # Conditioned Objs (%) | | | | |
| --- | --- | --- | --- | --- | --- |
| | 0 | 1 | 2 | 3 | 4 |
| Built-In Collision | 100.0 | - | - | - | - |
| SceneCollisionNet | 67.0 | - | - | - | - |
| LACO | 82.9 | 78.9 | 77.18 | 82.6 | 72.2 |

We evaluate the performance of LACO across different number of objects included in the language condition in Table 1. To measure collision prediction accuracy, we sample 10 trajectories with in total $N \approx 2000$ states, and evaluate $1/N \sum_{i=1}^{N} [y^l_{(i)} = \mathbb{1}_{C(o_i, s_i, l_i) > 0.5}]$. We find that LACO is robust to different numbers of conditioned objects, though it performs best when not conditioned on any objects. In this special case, LACO becomes a typical collision checker, without need to understand the semantics of the environment objects.

Even in the unconditional case, our method outperforms SceneCollisionNet [21], a point-cloud-based learned collision function. We use the official implementation of SceneCollisionNet, which is trained on a different simulator dataset. This distribution shift may be a reason for its poor performance in our environment. While our primary contribution is presenting a new paradigm of collision checking and path planning, the accuracy of our RGB-only method shows promise of collision checking without extensive camera setups and point clouds.

## 4.4 Generalization Experiments

One advantage of LACO is its ability to be modulated by language, which is flexible, abstract, and simple. We may expect that with its pretrained vision-language backbone, LACO may generalize to objects and language that are unseen in the training dataset.

**Generalization to unseen language.** We first evaluate generalization to unseen instructions of the seen objects. We compare the language-conditioned collision prediction accuracy with the original object name (**Default**), unseen synonyms of the object (**Synonym**), complex phrases describing the object (**Description**), and correct and incorrect colors (**Color**). For example, a collision to "hat" (Default) is evaluated with "beanie" for Synonym, "head-covering accessory" for Description, and "blue hat" for Color. For Color, we also use incorrect colors, e.g., "white hat", which ask LACO to ignore the objects. The exhaustive list of such variations can be found in Appendix, Table 9.

In Table 2, we find strong generalization for **Color**, where the language references the seen class name. LACO also generalizes to **Description**, while **Synonym** leads to 15.0% accuracy drop. One hypothesis is that the descriptions, which are typically longer and contain many keywords relating to the

Table 2: We evaluate generalization of LACO to instructions with unseen language.

| Default | Synonym | Description | Color |
|---|---|---|---|
| 78.9 | 63.9 | 71.4 | 77.0 |

object, may be more informative than just a synonym. Furthermore, short language conditions, like "cap," may even be ambiguous, as "cap" may refer to a bottle cap, a hat, or more. This suggests promise in future work of exploring stronger and more descriptive annotations, as we limit our language conditions to ShapeNet class names.

**Generalization to unseen objects.** We evaluate generalization to unseen objects across seen and unseen classes by measuring the collision prediction accuracy for language-conditioning on a single unseen object.

Table 3: Evaluation of LACO's generalization to new objects.

| Seen Class | | Unseen Class |
|---|---|---|
| Seen Object | Unseen Object | Unseen Object |
| 78.9 | 70.6 | 54.7 |

LACO achieves comparable collision prediction accuracy for unseen objects from seen classes, showing its strong generalization due to the pretrained vision encoder. However, LACO shows 24.2% lower accuracy for objects from unseen classes. Unlike the generalization to unseen language alone in Table 2, generalization to the unseen classes is more challenging as it has both class names and objects unseen during training.

## 4.5 Ablation Studies

**Pretrained observation encoder.** We investigate the benefit of using pretrained encoder by comparing a CLIP pretrained vision encoder and a CLIP vision encoder trained from scratch. In Table 4, we find that a pretrained CLIP encoder consistently outperforms the one trained from scratch.

Table 4: Ablation on Single and Multi-View Encoders.

| Method | Accuracy Per # Conditioned Objs (%) | | | | |
|---|---|---|---|---|---|
| | 0 | 1 | 2 | 3 | 4 |
| LACO | 82.9 | 78.9 | 77.2 | 82.6 | 72.2 |
| Finetuning | 71.5 | 64.9 | 65.14 | 72.9 | 68.9 |
| From scratch | 82.1 | 72.7 | 77.7 | 75.3 | 71.1 |
| LACO + MV | 68.8 | 69.4 | 76.3 | 73.4 | 68.9 |
| Finetuning + MV | 74.2 | 66.4 | 62.6 | 75.6 | 68.9 |
| From scratch + MV | 66.0 | 67.9 | 67.1 | 78.6 | 72.7 |

**Multiview camera inputs.** To extend to multi-camera RGB observations, we train a multi-view MAE to replace the CLIP vision encoder. The multi-view MAE is trained end-to-end to predict image reconstructions of two fixed camera views. Unlike MV-MAE [32] and Multi-MAE [33], we keep the original MAE masking ratio of 80% per each view. When objects are entirely masked out from one view, we find that they can be reconstructed if present in the second view. Sample reconstructions and hyperparameters are included in Appendix C. In Table 4, we find that using a pretrained MV-MAE, whether frozen or with finetuning, outperforms training from scratch. However, multi-view feature extraction remains an open problem, as single-view features lead to stronger predictions.

**Dataset size.** Although Section 4.4 shows the generalization capability of LACO, its generalization to unseen classes of objects is This may arise because we are training on a limited dataset of objects. As the quality and quantity of 3D assets increases, we may expect improved performance by training on more than a limited set of classes. We verify our hypothesis by

Table 5: The dataset size ablation examines how performance varies as we alter the size of the dataset.

| Dataset Size | Accuracy Per # Conditioned Objs (%) | | | | |
|---|---|---|---|---|---|
| | 0 | 1 | 2 | 3 | 4 |
| 50% | 82.9 | 74.2 | 73.4 | 78.0 | 73.6 |
| 80% | 81.2 | 70.0 | 72.6 | 72.2 | 65.8 |
| 100% | 82.9 | 78.9 | 77.2 | 82.6 | 72.2 |

varying the dataset size in Table 5. The results show that there is an improvement with the increased size of the dataset, but the improvement is marginal.

### 4.6 Language-Conditioned Path Planning Demonstrations

In this section, we showcase three tasks using language-conditioned path planning with LACO:

- **Reach Target (No Lang):** This task resembles a traditional path planning task, which aims to reach a target joint pose while avoiding obstacles. We do not condition on language.

- **Reach Target (1 Lang):** The objective is likewise to reach a target; however, 1 object is specified as collidable, allowing for more flexibility in plans.

- **Push Object (1 Lang):** The objective is to push an object forward. For this task, collisions are in fact desired, showcasing the usefulness of LAPP.

We report the success rates of LAPP in Table 6. Each path planning is considered successful if a valid path is found and the path reaches a target or pushes an object without undesirable collisions. The push object task is particularly well-suited for language-conditioned path planning: TrajOpt can be initialized with a trajectory passing through the object and op-

Table 6: We evaluate the success rates of LAPP-TrajOpt on three path planning tasks.

| No Lang | 1 Lang | |
|---|---|---|
| Reach Target | Reach Target | Push Object |
| 7/10 | 8/10 | 9/10 |

timize collision constraints with other objects. Reaching targets and pushing objects can also be composed to perform more complex tasks, such as avoiding all obstacles before reaching the object the arm needs to push.

### 4.7 Real-World Experiments

We show real-world trajectories with LAPP-TrajOpt in Figure 5. LAPP-TrajOpt with LACO, pre-trained on simulator data and finetuned with real-world data, is able to find plans in cluttered environments, using the language condition to discover a path that would otherwise be regarded as a trajectory with collisions.

## 5 Limitations

While LAPP with LACO offers promising advancements in contact-aware path planning, there are several limitations that should be acknowledged:

**Lack of environment dynamics.** LACO does not explicitly consider environment dynamics. Once an object is hit, it may react by being pushed or knocked down, potentially affecting the configuration and positions of other nearby objects. This restricts the ability of LAPP to handle dynamic environments and may lead to suboptimal or unsafe path plans when objects move significantly.

**Limited language prompt scope.** In our experiments, the language prompt is limited to specifying objects that are desirable or safe to collide. While this provides valuable control over contact conditions, the current scope of language prompts may not cover the full range of instructions

**Can collide with Pringles and Cheez-It box.**

**Can collide with Pringles.**

**Can collide with spam.**

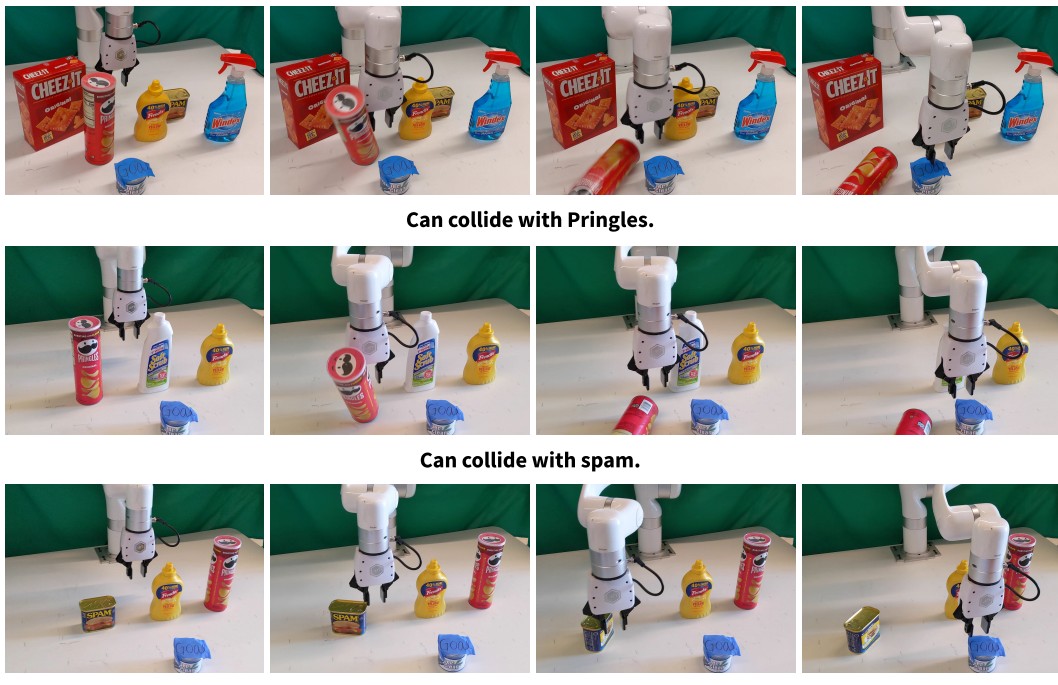

Figure 5: We demonstrate successful execution of the LAPP-TrajOpt path planning algorithm in the real-robot system. A path to the goal, specified by the blue jar, is blocked by diverse objects on the table, so the robot needs to collide with some object. We inform which object is safe and desired to collide using the language prompts: "pringles, cheezit", "pringles", and "spam", respectively.

or interactions that a user may desire. Including a wider variety of language instructions and specifications could enhance the versatility and adaptability of LAPP and LACO.

**Data generation efforts.** The training process for LACO relies on a combination of synthetic simulation data and manually collected real-world data. Both require significant human engineering and labeling efforts. Exploring advances in 3D asset availability [34] and simulation-to-real techniques [35, 36, 37, 38] could alleviate this limitation and enable more efficient training of LACO.

## 6    Conclusion

In conclusion, our proposed domain of Language-Conditioned Path Planning (LAPP) addresses the limitations of traditional collision-free path planning in contact-rich robotic manipulation tasks. By integrating contact awareness into path planning, LAPP allows robots to make informed decisions about contact with the environment, enabling them to perform complex manipulation tasks effectively.

As a first step towards LAPP, we propose to use Language-Conditioned Collision Functions (LACO), which learns to predict collisions based on visual inputs, language prompts, and robot configurations. This learned collision function eliminates the need for manual object annotations, point cloud data, or ground-truth object meshes, enabling flexible and adaptive path planning that incorporates both desired and controlled collisions.

**Acknowledgments**

We would like to thank Justin Kerr, Chung Min Kim, Younggyo Seo, and Hao Liu for their insightful advice on leveraging pretrained visual-language models and training transformer models. In addition, we thank Philipp Wu for assistance with the real-world experiments and Oleh Rybkin for feedback. This work was funded in part by Darpa RACER, Komatsu, and the BAIR Industrial Consortium.

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

# A Implementation Details

Table 7: LACO hyperparameters.

| Hyperparameter | Value |
|---|---|
| Learning rate | 3e-5 |
| Learning rate scheduler | cosine decay to 1e-7 |
| # Mini-batches | 32 |
| Training steps | 300000 |
| State tokenizer hidden units | (4096, 4096, 4096) |
| Prediction net hidden units | (512, 256) |
| Observation tokenizer/encoder | CLIP B/16 |
| Language tokenizer/encoder | CLIP B/16 |
| # Attention layers | 4 |
| # Attention heads | 16 |
| # Token dimension | 768 |

Table 8: TrajOpt hyperparameters.

| Hyperparameter | Value |
|---|---|
| # Steps | 10 |
| Velocity constraint | [-0.4, 0.4] |
| $\mu_0$ | 2 |
| $s_0$ | 0.01 |
| $c$ | 0.75 |
| $\tau^+$ | 1.1 |
| $\tau^-$ | 0.5 |
| $k$ | 10 |
| $f_{tol}$ | 0.0001 |
| $x_{tol}$ | 0.0001 |
| $c_{tol}$ | 0.01 |
| Solver | ECOS |
| # Penalty iterations | 5 |
| # Convexify iterations | 5 |
| # Trust iterations | 2 |
| Min. trust box size | 0.0001 |

# B Language Prompts for Evaluation

Language prompts are included in Table 9.

Table 9: Language prompts for evaluation.

| Original Noun | Synonym | Description |
|---|---|---|
| planter | plant stand | bin for plants |
| cap | hat, snapback | head-covering accessory, head-covering article of clothing |
| boat | sailboat, cruise ship | oceanic vehicle |
| yachting cap | hat, sailor hat | head-covering accessory |
| airplane | aircraft, airline | aerial vehicle, object that takes flight |
| omnibus | vehicle | long vehicle for travel, toy with wheels |
| bottle | water bottle, bottle container | container for fluids, travel-sized water container |
| flat cap | hat, beanie | head-covering accessory |
| laptop computer | electronic device, laptop | device for accessing internet, typing device for work |
| **Colors** | red, yellow, blue, purple, pink, gray, black, white | |

## C   Multi-View MAE

In addition to single-view observations, we also experiment with multi-view observations. Instead of using the pre-trained CLIP encoder, we pretrain a multi-view MAE from scratch on the simulator images. Then, we use multi-view features from the frozen multi-view MAE model for collision prediction.

In particular, we do not apply any special masking strategies. Though recent works in multi-view MAEs [33, 32] have applied such strategies, we find that even a basic MAE strategy leads to good reconstruction, even of parts occluded in just one view. For instance, in Figure 6, the blue object is completely masked out of the second view, yet the second view is able to successfully reconstruct the object.

We use an encoder with 2 layers and 16 heads, a decoder with 2 layers and 16 heads, token dimension of 768, patch size of 16, and masking ratio of 80%. Our image is preprocessed to be (224, 224), following convention. The learning rate is $3 \cdot 10^{-5}$. We add learned embeddings to the tokens for each view.

The result in Table 4 shows that multi-view LACO is worse than single-view LACO. We hypothesize that the use of pre-trained CLIP encoder is crucial for extracting useful features for collision prediction. However, we believe if the multi-view MAE is trained with large web-scale data, it can outperform single-view LACO.

# Multi-View MAE Training

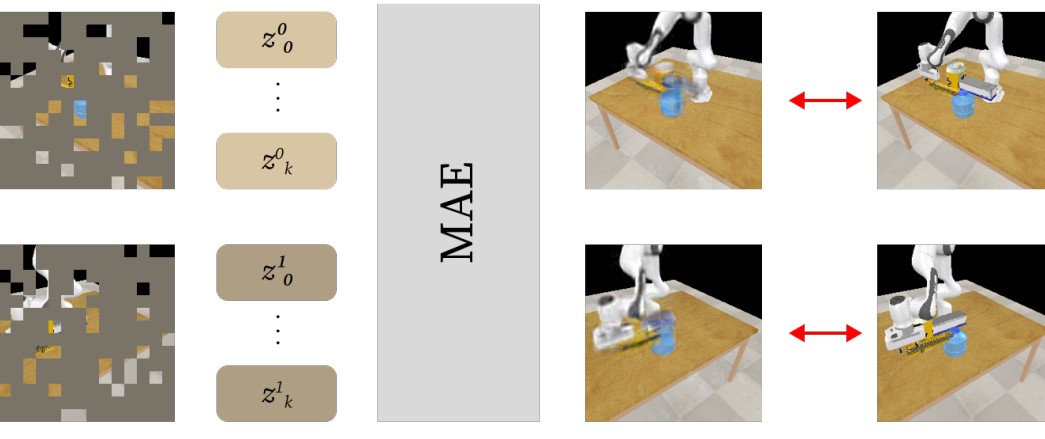

Figure 6: We pretrain a multi-view MAE that is trained from scratch on a simulator images. $k$ is the number of unmasked tokens passed into each view. Note that the blue object is completely masked out of the bottom view, yet it is able to be reconstructed.

