# OpenReview forum: "Language-Conditioned Path Planning"
_robot-learning.org/CoRL/2023/Conference — CoRL 2023 Poster_

### Official Review · Reviewer_CVWw · 2023-07-05

**Confidence:** 3
**Originality:** Good
**Technical Quality:** Very Good
**Clarity Of Presentation:** Very Good
**Impact:** 3

**Recommendation:**

Strong Accept: I recommend accepting the paper and will argue for my recommendation even if other reviewers hold a different opinion.

**Review:**

Strengths:

+ the considered problem of allowing collisions with certain semantic objects is quite interesting and practical.

+ The proposed method can be easily integrated with existing planners.

+ The provided experiments nicely demonstrate the proposed method.

Weakness

- The technical approach needs to be motivated more. Existing semantic planners can possibly be used to address similar problems without the high computational cost of designing neural network-based collision functions.



**Quality Of The Limitations Section:**

Limitations are addressed clearly

**Questions For Rebuttal:**

Questions

1) Why a neural network model (LACO) is needed to check for collisions given the current robot configurations? The same can possibly done using existing semantic planners such as [A]-[C]. Such planners reason about the location and semantic labels of nearby objects and based on that they make decisions about what objects should be approached and what objects need to be avoided.

[A] "Perception-based temporal logic planning in uncertain semantic maps." IEEE Transactions on Robotics 38.4 (2022): 2536-2556.

[B]  "Reactive semantic planning in unexplored semantic environments using deep perceptual feedback." IEEE Robotics and Automation Letters 5.3 (2020): 4455-4462

[C] "Sscnav: Confidence-aware semantic scene completion for visual semantic navigation." 2021 IEEE international conference on robotics and automation (ICRA). IEEE, 2021

The authors do not need to cite these specific papers; however, a discussion on how this work differs from existing semantic planners would be helpful.

2) What do the numbers in Table 6 mean?

3) How does LACO generalizes to unseen environmental conditions? For example, consider seen objects/classes but different lightning conditions. A discussion on this would be helpful.

**Robotics Focus:**

Sufficient demonstration on hardware

**Summary Of Paper:**

This paper presents a novel collision checking function that can seamlessly integrated with existing planners. The novelty of this function is that it considers cases where collision with certain objects is allowed. Acceptable collisions are specified as a language prompt. Then proposed collision checking function, called LACO, reasons about collision using a single image, a language prompt, and the current robot configuration.





**Summary Of Recommendation:**

Overall the paper addresses an interesting problem and the proposed algorithm can be a valuable contribution in the field of robot planning.

---

### Official Review · Reviewer_FRPN · 2023-07-18

**Confidence:** 3
**Originality:** Excellent
**Technical Quality:** Good
**Clarity Of Presentation:** Very Good
**Impact:** 3

**Recommendation:**

Weak Accept: I recommend accepting the paper, but will not argue for my recommendation if the majority of other reviewers have a different opinion.

**Review:**

The authors introduce a new robotics domain where certain collisions are desired while others are not. They propose the use of language prompts to express human ideas effectively. The design of a language conditioned collision checker is novel and inspiring. The writing in the paper is clear, and the methods are well-presented. Considering that language conditioned planning is a new problem, it is acceptable that there isn't extensive comparison demonstrated in the paper. The limitation section highlights some drawbacks of the current methodology and suggests that making the language prompts more general would enhance the significance of the work.

**Quality Of The Limitations Section:**

Limitations are addressed clearly

**Questions For Rebuttal:**

There is a concern regarding whether a single image representing the scene is sufficient for the problem. Since geometry information cannot be obtained from the image, the collision prediction by the network may be inaccurate. It would be helpful if the authors could elaborate on whether the accuracy could be improved by incorporating point clouds or multi-view images into the network.

**Robotics Focus:**

Sufficient demonstration on hardware

**Summary Of Paper:**

The paper proposes a path planning algorithm that includes a collision function based on language conditions. The function calculates a collision cost based on the given image, robot state, and language prompt. This collision function, called the language conditioned collision function, outputs a collision score. By incorporating this function into a sampling-based planning algorithm, the robot can navigate to the goal while considering desired or undesired collisions. The authors train the network using both simulation and real-world data. The paper also discusses the advantages over existing collision checkers and presents some ablations.

**Summary Of Recommendation:**

The paper provides valuable insights into the concept that collisions can be desirable in robot manipulation tasks. To differentiate between desired and undesired collisions, the paper introduces a language-conditioned collision checker. By utilizing language prompts, the robot can plan a path that includes acceptable collisions with specific objects. Real-world demonstrations validate the effectiveness of the method. Overall, the paper showcases an implementation that contributes to the development of higher-level intelligent robots.

Update: Thanks for the efforts of the authors. My concern is addressed. I recommend accepting this paper.

---

### Official Review · Reviewer_bXSy · 2023-07-18

**Confidence:** 4
**Originality:** Very Good
**Technical Quality:** Very Good
**Clarity Of Presentation:** Very Good
**Impact:** 4

**Recommendation:**

Weak Accept: I recommend accepting the paper, but will not argue for my recommendation if the majority of other reviewers have a different opinion.

**Review:**

The heart of this paper is the interesting viewpoint that, in many motion planning settings, not all collisions are created equal. In allowing some collisions, the planner can solve previously impossible or very difficult tasks. The paper focuses on the setting where the collision information is provided via language conditioning (LAPP). There is a wide range of semantic information that one could imagine extracting from language, i.e. that is okay to partially collide with an object like a shower curtain since the curtain will confirm, that colliding with a full glass of water is more disastrous than an empty glass, or that for some objects it is okay to slightly bump the object as long as that contact is minor. This is exciting and I believe future work could build on the idea of LAPP to explore these types of problems.

As a step towards this, the paper proposes the more narrow, although very well-defined, application of using language to specify which objects to collision check against (LACO). LACO's methodology is well-described and it leverages state-of-the-art work in encoding images and language. The specification is such that it could be easily plugged into a motion planner (as the paper demonstrates).

One major question is that it is not clear what the syntax/form/phrasing of the language command is. Are the descriptions given in Fig. 5 examples of the language command or is the language command the list of objects? Does the syntax impact performance? Additionally, LACO outputs collision probability. While this is converted to a binary value (I believe through thresholding), it would be interesting to consider treating it as a cost in the context of an optimization-based motion planner.

The experiments evaluate LACO as compared to non-language conditioned planners and conduct several ablation studies. A variant of TrajOpt where LACO is used as the collision-checker is applied to a reaching and pushing task. For the pushing task, there are only three instances and the results are the same with and without conditioning on language, which does not support the claim that conditioning on language is useful in this task. Additionally, how is the downstream task success defined? For the failures, does this mean that there was an undesirable collision, no path was found or that the robot failed to reach the target?

The paper discusses several interesting limitations that serve as great areas of future work. In general, the paper is well-written and easy to follow. The related work seems sufficient.

A few more minor questions and comments:
- Does the single-view image include the robot? Sec 3.2 states that the environment configuration is agnostic to the robot joint states - is the robot masked out of the image?
- Sec 4.2 mentions that state is respected to (x, y) position of the end effector. One interpretation (that seems unlikely) is that only the end effector is considered for collisions, not the entire configuration is the arm. Since I think this is the wrong conclusion, what does state mean in this case? Is it that, in collecting real world data, the objects and end effector position were only varied in (x, y)?
- Since SceneCollisionNet is trained on a different data set, its unclear if the comparison is fair. However, I do not see this as a major concern, since SceneCollisionNet is not solving the same problem as LACO and the comparison is only one small part of the results.
- For the simulation results, is the Built-In collision checker used as ground truth?
- Sec 3.3, Line 134: "implement LAPP using a sampling-based method, LAPP-TrajOpt [7]." -> "implement LAPP using an optimization-based method, LAPP-TrajOpt [7]."
- Sec 3.3, Line 135: "additionally integrate a sampling base method, LAPP-RRT* [6]" -> "additionally integrate a sampling-based method, LAPP-RRT* [6]"

**Quality Of The Limitations Section:**

Limitations are addressed clearly

**Questions For Rebuttal:**

- What is the form of the language command?
- LACO outputs a collision probability. Is this value then thresholded when evaluating performance and when used within a motion planner? (If so, what is that threshold?)
- What are the typical failure cases for LACO?
- Sec 4.4 considers generalization to unseen language by considering synonyms and descriptions. How were these synonyms and descriptions sourced?
- In this context of the reaching task for Sec 4.5, how is conditioning on language helpful? In the current results the performance is the same, with or without language.

I have included a few more minor questions in the "Review" section. Additionally, if there are videos of the real world experiments, it would be great to include them in the supplemental.

**Robotics Focus:**

Sufficient demonstration on hardware

**Summary Of Paper:**

The paper proposes relaxing the collision-free requirement of motion planning via the introduction of "Language-Conditioned Path Planning" (LAPP). In LAPP, the goal is to generate a joint-space path where semantic information (as a language command) decides what collisions are allowed. The paper proposes Language-Conditioned Collision Function (LACO), which takes in a single-view image, a robot configuration and a language command and returns a collision probability.

Trained with cross-entropy loss, the learning pipeline tokenizes the image, language and robot configuration by a CLIP vision encoder, CLIP language model and MLP respectively. The tokens are passed into a transformer and the output is averaged to provide collision probability. The collision function is trained with a large simulation dataset, fine-tuned with real-world data. Ablation results compare the LACO's performance and the function is also integrated into an optimization-based motion planner.

**Summary Of Recommendation:**

The paper presents an interesting take on modulating what it means for a path to be collision-free. The method incorporates advances from computer vision and natural language and incorporates them into a motion planning setting. There are extensive ablation studies and motion planning experiments. There are a number of request clarifications that would greatly strengthen the paper. However, I believe these could be feasibly addressed during the rebuttal/revision period.

[After the rebuttal, I would still recommend the paper for acceptance. I have also added a comment below]

---

### Author Response · Authors · 2023-08-11
**Global Response**

We thank all reviewers for their constructive feedback and for helping us make our paper a stronger submission!

We updated our paper according to the reviewers’ suggestions and included new experimental results. Please check our updated paper attached to each rebuttal.

We hope we have addressed all your concerns and questions. Please let us know if there are any concerns preventing you from raising your score.

---

### Author Response · Authors · 2023-08-16
**Area Chair Rebuttal Summary**

Dear Area Chair,

We sincerely appreciate your time and effort in serving the CoRL community.

As reviewers highlighted, our project introduces a novel and practical new paradigm of language-conditioned path planning (all reviewers), combining advances in computer vision and natural language (Reviewer bXSy), with clear and strong experimental results (Reveiwers FRPN and CVWw).

We are glad to find all reviewers are very positive about the paper (3 Weak Accept), with Reviewer CVWw recommending the paper for acceptance.

Unfortunately, after posting our response, we have _not_ heard back from the remaining reviewers FRPN and bXSy. We have provided many additional experiments, clarifications, and real-world videos. We sincerely believe that this successfully addresses all concerns of the reviewers and that these clarifications and additional results further strengthen our paper.

Thank you very much!
Authors

---

### Decision · Program_Chairs · 2023-08-30

**Decision:**

Accept (Poster)

**Comment:**

This paper suggests a new approach called "Language-Conditioned Path Planning" (LAPP) that relaxes the need for collision-free motion planning. LAPP works by allowing semantic information, like language commands, to determine which collisions are acceptable as it generates a path in the joint space. The paper presents an interesting, novel approach backed by extensive ablation studies and motion planning experiments.